# Effect of Small Extracellular Vesicles Produced by Mesenchymal Stem Cells on 5xFAD Mice Hippocampal Cultures

**DOI:** 10.3390/ijms26094026

**Published:** 2025-04-24

**Authors:** Daria Y. Zhdanova, Natalia V. Bobkova, Alina V. Chaplygina, Elena V. Svirshchevskaya, Rimma A. Poltavtseva, Anastasia A. Vodennikova, Vasiliy S. Chernyshev, Gennadiy T. Sukhikh

**Affiliations:** 1Institute of Cell Biophysics, Federal Research Center Pushchino Research Center for Biological Studies, Russian Academy of Sciences, Institutskaya 3, Pushchino, 142290 Moscow, Russiashadowhao@yandex.ru (A.V.C.); 2Shemyakin-Ovchinnikov Institute of Bioorganic Chemistry, Russian Academy of Sciences, Ulitsa M0iklukho-Maklaya 16/10, 117997 Moscow, Russia; wodemikanasty@mail.ru; 3National Medical Research Center for Obstetrics, Gynecology and Perinatology Named After Academician V. I. Kulakov, Ministry of Healthcare of the Russian Federation, Oparina St. 4, 117997 Moscow, Russia; rimpol@mail.ru (R.A.P.); v_chernyshev@oparina4.ru (V.S.C.); gtsukhikh@mail.ru (G.T.S.); 4Institute of Bioorganic Chemistry, National Research Nuclear University “MEPhI”, Kashirskoe Shosse 31, 115409 Moscow, Russia; 5Skolkovo Institute of Science and Technology, Bolshoy Boulevard 30, Bld. 1, 121205 Moscow, Russia

**Keywords:** small extracellular vesicles, 5xFAD, Alzheimer’s disease, beta-amyloid, luminex, tetraspanins

## Abstract

Alzheimer’s disease (AD) is one of the most common progressive neurodegenerative diseases leading to impairments in memory, orientation, and behavior. However, significant work is still needed to fully understand the progression of such disease and develop novel therapeutic agents for AD prevention and treatment. Small extracellular vesicles (sEVs) have received attention in recent years due to their potential therapeutic effects on AD. The aim of this study was to determine the potential effect of sEVs in an in vitro model of AD. sEVs were isolated from human Wharton’s jelly mesenchymal stem cells (MSCs) by asymmetric depth filtration, a method developed recently by us. AD was modeled in vitro using cells obtained from the hippocampi of newborn 5xFAD transgenic mice carrying mutations involved in familial AD. After isolation, sEVs underwent detailed characterization that included scanning electron microscopy, nanoparticle tracking analysis, confocal microscopy, Western blotting, and Luminex assay. When added to 5xFAD hippocampal cells, sEVs were nontoxic, colocalized with neurons and astrocytes, decreased the level of Aβ peptide, and increased the synaptic density. These results support the possibility that sEVs can improve brain cell function during aging, decrease the risk of AD, and potentially be used for AD therapeutics.

## 1. Introduction

Alzheimer’s disease (AD) is characterized by the development of dementia with memory loss and a significant deterioration in cognitive abilities [1]. In the absence of effective treatments for AD, by 2050, the number of cases in the world is expected to reach more than 100 million people [2]. So far, only two categories of drug treatments are approved for AD: cholinesterase inhibitors (tacrine, donepezil, rivastigmine, and galantamine) and a partial N-methyl-D-aspartate (NMDA) receptor antagonist (memantine), which only to some extent slows down the progression of the disease.

Aggregation and deposition of β-amyloid (Aβ) in the brain leads to neuronal dysfunction and is used as the most common target of drugs being developed [2,3,4,5,6]. However, the role of Aβ in the genesis of AD is not completely accepted due to the lack of clinically significant efficacy of anti-Aβ therapy [7]. It is hypothesized that the amyloid pathology manifests early at asymptomatic stages of AD. If this is the case, anti-Aβ therapy may be more effective at early stages of AD. The low efficacy of anti-Aβ monotherapy stimulates a search for new drugs such as antidepressants, anticonvulsants, antipsychotics, antioxidants, anti-inflammatory drugs, and others [7,8,9,10]. As an alternative, cell therapy can be used for AD treatment [11,12,13,14]. Previously, the therapeutic effect of mesenchymal stem cells (MSCs) was shown in AD models [15,16]. Such therapy resulted in a decrease of Aβ plaques, neuroinflammation, cholinergic dysfunction, and pericytic and synaptic loss, providing immunomodulatory and neuroprotective effects [15]. However, despite the positive effects of MSCs, their application is traumatic, associated with the risk of thrombosis, as well as with problems with blood–brain barrier (BBB) transit.

Intranasal (i.n.) administration of drugs is a non-invasive, easily accessible approach for drug delivery to the brain. However, MSCs are too large for i.n. administration. An alternative to MSCs are extracellular vesicles (EVs) secreted by these cells [17]. Among EVs, sEVs that include exosomes are the most characterized and studied. Exosomes are vesicles with a hydrodynamic diameter of 30–200 nm, mostly originating from endosomal compartment of cells and containing cargo in the form of proteins, lipids, RNA, and DNA [18,19,20,21]. A number of studies have shown that MSC-derived sEVs reduce cognitive impairment in various animal models of neurological disorders, including AD [20,21,22]. sEVs induce a synergistic effect on metabolism, neuroinflammation, migration of cell precursors, and angio-, neuro-, and synaptogenesis. Previously, we demonstrated in a mouse model of sporadic AD that the restoration of spatial memory can be achieved through i.n. administration of sEVs derived from Wharton’s jelly MSCs [22]. In this presented work, we investigate the effects of MSC-derived sEVs on hippocampal cell cultures obtained from newborn transgenic (Tg) 5xFAD mice, carrying the genotype with five mutations linked to an increased production of Aβ and serving as a model for hereditary AD.

## 2. Results

### 2.1. Characterization of sEVs

To obtain sEVs, MSC culture supernatants were purified by asymmetric depth filtration from 6 cultures obtained from different women. The average concentration of sEVs was determined by nanoparticle tracking analysis (NTA) and was found to be 4 × 10^10^ particles/mL. Transmission electron microscopy (TEM) and NTA analysis demonstrated the round shape and heterogeneity in size of isolated sEVs (Figure 1a,d) with an average diameter of 118 ± 21 nm, with two observed peaks, 80 and 150 nm (Figure 1b). Judging from the clear heterogeneity, there is a possibility that these sEV fractions are generated by different mechanisms. The larger particles are likely to be ectosomes produced by cytoplasmic membrane budding [23], while the smaller ones can be exosomes released after the fusion of multivesicular bodies with the plasma membrane. All sEV fractions were used in this study. The sEV ζ-potential was found to be −10.8 ± 0.4 mV.

Purified sEVs contained different proteins, including soluble proteins such as growth factors, chemokines, and cytokines (Figure 1b,c, Appendix A). Among the 41 soluble proteins detected by multiplex assay, the MSC supernatant contained detectable levels of 30 factors (Appendix A). Cytokines TGFa, IL-12P70, IL-13, IL-15, IL-17A, IL-9, IL-1b, IL-2, IL-3, IL-5, and TNF-b were low or completely undetected in sEVs. There was a high variability in humoral factor content in different sEV samples. The concentration of the chemokines MCP1, GRO, and fractalkine and growth factors FGF2 and VEGF was approximately 100 pg/mL in sEVs (Figure 1b), while the level of cytokines was tenfold smaller (Figure 1c). Pearson’s correlation analysis demonstrated that the concentration of the soluble factors in the flow-through supernatants after sEV purification and in sEVs correlated significantly (r = 0.504, *p* < 0.05) (Appendix A). These results demonstrate that sEV cargo reflects the content of cells which produce them. According to the statement from the International Society for Extracellular Vesicles [23], sEVs must be positive for membrane-bound proteins such as CD9, CD63, and CD81, contain a lipid layer and intracellular proteins such as heat shock proteins, and must be negative for the proteins associated with other intracellular compartments such as calnexin. Western blotting analysis demonstrated that MSC sEVs express CD9, are positive for HSP70, and are negative for calnexin (Figure 1f). All these proteins were detected in control cells (A431). Multiple other protein bands were detected both in cells and sEVs (Figure 1f). The lipid content of sEVs was analyzed using lipid dye BDP.

### 2.2. Estimation of sEV Cytotoxicity in Hippocampal Cultures

The evaluation of sEVs cytotoxicity in the primary culture of 5xFAD mouse hippocampi showed that the introduction of sEVs at doses ranging from 4 × 10^6^ to 160 × 10^6^ sEVs increased the survival rate of 5xFAD cells (Figure 2). The effect was dose-dependent, reaching a saturation level at high doses of sEVs (*p* < 0.05).

### 2.3. Interaction of sEVs with Tg and Non-Transgenic Hippocampal Cells

The interaction of sEVs with neurons and astrocytes was analyzed using the cell hippocampal cultures from both transgenic and non-transgenic mice. For visualization, sEVs were stained with lipid dye Dil. Neurons and astrocytes were detected using antibodies to MAP2 or GFAP markers, accordingly. The results show that sEVs colocalize with both neurons and astrocytes in nTg and Tg cultures (Figure 3).

To determine with which cells of the neuroglial culture sEVs interact with to a greater extent, we analyzed the ratio of the average intensity of red fluorescence (MFI) normalized to green MIF, corresponding to the immunopositive staining of neurons or glia, respectively, in Tg and nTg cultures. To test for a statistical difference, areas with a predominance of neurons or astroglia were identified, and five to eight microscopic areas in each repeated experiment were scanned and analyzed. The red/green ratio in the Tg culture was significantly higher than in the nTg culture (Figure 3m), showing that colocalization of sEVs with neurons and glia in transgenic 5xFAD cells is higher in Tg compared to nTg.

### 2.4. Effect of sEVs on the Expression of Aβ

To study the effect of sEVs on Aβ expression in Tg and nTg cells, different quantities (0, 4 × 10^6^, 40 × 10^6^, 80 × 10^6^, or 160 × 10^6^) of sEVs were used. Cells were cultivated with sEVs for 24 h and stained for Aβ 1–42 peptide (Figure 4). Control Tg but not nTg cells expressed Aβ 1–42. Cultivation of Tg hippocampal cells with sEVs resulted in a decrease in Aβ fluorescence (Figure 4a–e). Analysis of Aβ immunopositive areas in Tg cultures showed a dose-dependent effect, with its maximum being at high sEV concentrations (≥80 × 10^6^) (Figure 4g).

### 2.5. Expression of Synaptophysin in Tg Cells

The effect of sEVs on the synaptic density was analyzed in Tg cultures by fluorescent microscopy using antibodies to synaptophysin (SP) and MAP2. As a result of immunocytochemical analysis, sEVs increased the synaptic density in Tg cultures compared to the control Tg untreated ones (*p* < 0.01), mostly at a high concentration of sEVs (80 × 10^6^) (Figure 5).

## 3. Discussion

Mechanisms of sEV activity are currently poorly understood and are under intensive investigation in experimental and clinical studies [24,25]. Due to sEV heterogeneity, where sEVs contain different types of cargo, and the need to keep markers within the lipid bilayer for them to be active, the progress in determining all types of sEV mechanisms is slow. Proteomics, RNA-seq, and miR-seq demonstrate that more than a thousand types of biomolecules are carried by sEVs [26]. sEV content varies and depends on the source used for purification, however, high overlaps can be observed [24]. Up to 70–90% of identified proteins originate from the cell cytoplasm, which supports our findings. Still, multiple works demonstrated that sEVs from stem cells may participate in regeneration and normal physiology, while those derived from infected or cancer cells mediate pathogenesis [25,27,28,29,30,31]. Based on our previous unpublished work on aging cell cultures, we hypothesize that sEV turnover is a homeostatic process in which aging cells deliver valuable cargo to the surrounding young cells to support the health of the tissue and stimulate normal functioning of the cells. In this regard, sEVs delivered to neuronal cells can improve homeostasis of the brain.

Numerous studies are ongoing in search for an effective AD treatment. Both in vitro and in vivo AD models demonstrate specific changes in brain cells. The characteristic signature of AD is an increased level of Aβ aggregates in the brain, leading to neuronal death [1,2]. In this study, we used hippocampal cell cultures obtained from 5xFAD mice co-expressing AD-specific mutations in amyloid precursor protein (APP) and presenilin I as an in vitro AD model. These 5xFAD mice are characterized by Aβ expression in their brains, synaptic markers synaptophysin and syntaxin and postsynaptic density decreases, and neuron loss [32]. Previously, authors demonstrated Aβ expression, Tau protein phosphorylation, and the suppression of genes associated with learning, memory, and synaptic plasticity, including the gene of brain neurotrophic factor (exon Bdnf IV) and Homer1, on a human nerve cell culture that consisted of a neuroblastoma cell line overexpressing human APP with FAD double mutations [33]. Several other in vitro models of AD were presented [32,33,34,35]. Here, we clearly defined immunopositivity for Aβ in the 14-day culture of hippocampal cells from 5xFAD mice. Aβ immunopositivity is absent in the hippocampal culture of wild type nTg mice.

Multiple publications demonstrate sEV effects, both using in vivo and in vitro models, on AD-specific features [36,37,38]. Chen et al. used the human neuroblastoma cell line SH-SY5Y, expressing the same APP and presenilin I mutations (K670N/M671L and V717I) specific to familial AD, and incubated differentiated cells with sEVs from Wharton’s jelly MSCs for one week [39]. A decrease in the level of Aβ and the restoration of the expression of neuronal memory and synaptic plasticity genes were found after the sEV treatment. In this study, we demonstrated that only 24 h after the administration of sEVs are needed to see a significant decrease in the immunopositive response to Aβ 1–42 when compared with control samples. The effect of sEVs was dose-dependent; however, even low doses showed effectiveness. Earlier, we showed that sEVs isolated from the human Wharton’s jelly MSCs are applicable for treatment of neurodegenerative disorders in vivo in a model of sporadic AD in mice [22].

Additionally, as expected, sEVs are nontoxic. We observed an increase in Tg cell survival after sEVs co-cultivation. It can be hypothesized that the increase in cell viability can be a result of the decrease in the content of Aβ (1–42), and an increase in synaptic connections, as demonstrated by an increased synaptophysin level in neurons. Although the exact function of synaptophysin is not completely understood, it plays a role in the formation of the synaptic contacts. Loss of synaptophysin was found in AD, but more evidently at high hippocampal taupathy [40]. Earlier, Xin et al. demonstrated in ischemic rats that systemic administration of sEVs results in a significant increase in density of axons and synaptophysin-positive sites in the affected areas of the cortex and striatum compared to untreated animals [41].

Another possible explanation of the regenerative effect of sEVs may be connected to the growth factors delivered by sEVs to the hippocampal cell culture. Some authors hypothesize that vesicle activity is mostly mediated by the protein content of sEVs [42,43,44]. In a cellular model of AD, Katsuda et al. showed that the transfer of sEVs from adipose tissue MSCs to N2A neuroblastoma cells, expressing both human APP and presenilin 2 mutations, causes a decrease in the levels of secreted and intracellular Aβ peptide [45]. The authors hypothesized that sEVs derived from MSCs exhibit the activity of an enzyme specific to neprilysin, which is one of the most well-known enzymes involved in the degradation of Aβ peptide [46]. The activity of neprilysin is reduced in patients with AD [47]. We believe that our result may be related to the activity of this enzyme. Here, we quantitatively estimated the concentration of some proteins in sEVs by using multiplex Luminex technology. A total of 40 factors were analyzed. sEVs contained only eight factors: growth factors EGF, FGF-2, and PDGF, cytokine IL-1α, and chemokines RANTES, MDC, IP-10, and Eotaxin.

Among the compounds identified in sEVs that may exert a neuroprotective effect on cultured cells, several of them are of particular interest. FGF2 suppresses glutamate-induced cell death by reducing intracellular calcium accumulation, decreasing apolipoprotein secretion, inhibiting free radical accumulation, slowing Aβ-induced neurodegenerative cascades, and counteracting oxidative stress and mitochondrial dysfunction caused by Aβ peptides [48]. MCP-1 contributes to neural tissue regeneration by promoting differentiation and stimulating astrocytes and microglia to produce neurotrophic factors, including basic fibroblast growth factor (bFGF) [49]. Fractalkine (CX3CL1, FKN) is primarily expressed on neurons, while its receptor (CX3CR1) is exclusively found on microglia. This chemokine, unique for its receptor specificity, with a one-to-one relationship with its receptor, signals through its cognate receptor to reduce the expression of pro-inflammatory genes in activated microglia [50]. PDGF demonstrates neuroprotective action through the signaling pathways involved in oxidative stress responses [51]. VEGF exhibits neurotrophic, neuroprotective, and angiogenic properties. It directly inhibits programmed cell death and stimulates neurogenesis. VEGF also serves as a mediator in numerous processes, including antioxidant activation, which indirectly contributes to neuroprotection [52]. Although sEVs contain biologically active substances demonstrating individual properties, their therapeutic potential is determined by the coordinated physiological action of these bioactive compounds.

The question of whether sEVs colocalize with neuronal cells was of our particular interest, since neurons are the subject of the most pronounced destructive processes in AD. Bodart-Santos et al. studied the internalization of Wharton’s jelly MSC-derived sEVs by primary hippocampal cells from rat embryos [53]. Exposure of hippocampal cells to Aβ oligomers AβOs for 24 h, and a subsequent incubation with sEVs for 24 h, resulted in sEVs internalization by astrocytes but not by neurons. We found that sEVs colocalized not only with astrocytes, but with neurons as well. MSC sEVs internalization by both neuronal cell bodies and axons was observed in cultured primary cortical neurons [54]. Therefore, MSC sEVs can directly transfer their cargo to recipient neurons and thereby influence neuronal remodeling.

The effect of sEVs in in vitro AD models can depend on the sources of sEVs, type of cultures, number of sEVs, and time of co-incubation. Other molecules such as endopeptidases, antioxidant enzyme catalase, heat shock proteins, and glycosphingolipids are believed to be involved in Aβ cleavage [55,56,57,58,59,60,61,62]. However, in our studies, the possible influence of endogenous sEVs present in the astroneuronal culture should be taken into account, since in AD patients, the neurotoxic effect of astrocyte-derived sEVs is evident with the overlap of Aβ plaque density and initiation of microglial cytotoxic attacks damaging neurons [63,64,65].

The effects of microRNA present in exogenous sEVs and influencing neuronal plasticity recently received particular attention. Transfer miRNA stimulates parenchymal brain cells to secrete growth and trophic factors, which promote neurorestorative mechanisms such as angiogenesis, neurogenesis, synaptogenesis, oligodendrogenesis, and anti-inflammatory responses. In particular, miR-17-92 takes part in the mechanisms of the MSC sEV effect on axonal growth by activating the PTEN/mTOR signaling pathway in recipient neurons [54]. The neuroprotective and neurorestorative effects of sEVs may be due to the modulation of gene, protein, and miR expression in target cells and tissues [66,67]. miRs regulate post-transcriptional gene regulation by binding to complementary sites in the 3′-untranslated region of messenger RNAs [68,69].

Most of the internalized sEVs are used by the cell itself, either undergoing lysosomal degradation or recovery of some components via the trans-Golgi network [70]. However, some internalized sEVs are re-secreted together with endogenous sEVs and can perform a longer-distance action [71]. It is likely that MSC sEVs exert their therapeutic potential by acting directly on target cells, and also via secondary secretion of sEVs by astrocytes and other parenchymal cells. Perhaps, in our study, the observed stepwise dose-dependent effects of sEVs on various characteristics of the culture were related to the different fates of internalized sEVs. However, this interesting observation requires a further detailed investigation.

## 4. Materials and Methods

### 4.1. Mesenchymal Stromal Cells (MSCs)

The primary cultures of MSCs were developed from Wharton’s jelly of the umbilical cord tissue collected from six healthy women. Sampling was carried out with written informed consent of the women in labor after a cesarean section. Further isolation of MSCs was carried out in full accordance with the protocol #8 from 26 August 2021 (National Medical Research Center for Obstetrics, Gynecology and Perinatology named after Academician V.I. Kulakov). Cell cultivation was carried out in 5% CO_2_ at 37 °C in DMEM/F12 medium (1:1, Gibco, Waltham, MA, USA) supplemented with 10% fetal calf serum, 2 mM L-glutamine (cat. no 1.3.8.3), 100 U/mL penicillin, and 100 μg/mL streptomycin (cat. no. A065) (PanEco, Moscow, Russia) in 25 cm^2^ culture flasks (Corning, Corning, NY, USA). The medium was changed by 50% every three days. Upon reaching a state of 80% confluence, cells were seeded at a ratio of 1:2. Trypsin/EDTA solution (PanEco, Moscow, Russia, cat. no. PO39n) was used to remove cells from the plastic surface. The expression of MSC surface markers was analyzed using specific antibodies conjugated with phycoerythrin (PE) to CD90, RRID:AB_131291; CD105, RRID:AB_1575944; CD73, cat. no. B68176; CD19, cat. no. A07769; and HLA-DR, RRID:AB_131284 (all from Beckman Coulter, Brea, CA, USA).

### 4.2. Production and Characterization of the MSC-Derived sEVs

MSCs, after passages 2–4, were transported to serum-free DMEM/F12 medium and incubated for 48 h. The culture supernatant was centrifuged for 20 min at 3000× *g* to remove cell debris and microvesicle aggregates. The clarified culture liquid was used to isolate sEVs by asymmetric depth filtration [72] according to the manufacturer’s protocol (Prostagnost, Moscow, Russia). The obtained sEV samples were stored in protein LoBind tubes (Eppendorf, Hamburg, Germany) at −20 °C after isolation.

### 4.3. PAAG Gel-Electrophoresis and Western Blotting

A total of 100 µL of sEVs were lysed using RIPA buffer (Beijing Solarbio Science& Technology Co., Ltd., Beijing, China) in the presence of a cocktail of inhibitors (ab201119, Abcam). The protein concentration was determined by BCA assay (ThermoScientific, Waltham, MA, USA). A total of 30 μg protein was mixed with the loading buffer, denaturated for 10 min at 95 °C, and loaded onto 12% polyacrylamide gels to separate proteins. The proteins were then transferred to PVDF membranes (BioRad Laboratories, Hercules, CA, USA), and 5% skimmed milk was used to block the membranes at room temperature for 1 h. Subsequently, the membranes were incubated with primary antibodies to CD9, HSP70, or calnexin (AffiGEN, San Jose, CA, USA) at 4 °C overnight, washed out, and secondary anti-rabbit HRP or anti-mouse HRP antibodies (RRID: AB_2650737 and AB_315009, Biolegend, San Diego, CA, USA) were added at room temperature for 1 h. Protein bands were visualized using 3,3′-Diaminobenzidine tetrahydrochloride (DAB, Merck KGaA, Darmstadt, Germany) and a gel imaging system (Bio-Rad Laboratories, Inc., Hercules, CA, USA).

### 4.4. Transmission Electron Microscopy (TEM)

Prior to TEM, 10 µL of sEVs diluted 1:50 in DI water were deposited onto a clean copper grid and incubated for 1 min. After removal of excess liquid by blotting, 10 µL of 2% uranyl acetate (Merck, Darmstadt, Germany) were deposited and the sample was incubated for 30 s. Excess liquid was removed by blotting. Images were acquired using a Cs-corrected TEM instrument (FEI, Titan Themis Z, Hillsboro, OR, USA) at 200 kV. Illumination was performed with an e-beam spot size of 5 and probe current of 0.15 nA. The acquisition with an exposure time of 2 s was performed using a Ceta 2048 × 2048 bottom-mounted CCD camera and Velox software 3.13 (FEI Company, Hillsboro, OR, USA).

### 4.5. Nanoparticle Tracking Analysis (NTA)

Prior to NTA, sEV samples were thawed and diluted 1:100 in DPBS. NTA was performed using a Nanosight model NS-300 equipped with a 45 mW 488 nm laser (Malvern, Salisbury, UK). Within several minutes following dilution, the sample was injected into the test cell, illuminated by the laser, and 5 videos were recorded (60 s each) by a high sensitivity sCMOS camera (camera level set to 15). Approximately 30–40 particles were observed in the field of view during capture of the videos. After recording, videos were analyzed using the Nanosight software (version 3.2) with the detection threshold set to 5 to determine the concentration of sEVs and their size distribution. The viscosity of DPBS was assumed to be that of water at the temperature when measurements were performed. NTA data was used to calibrate sEVs by protein content (Appendix A).

### 4.6. ζ-Potential

Prior to determining the ζ-potential of the sEVs, the sample was diluted 1:10 in DPBS. A Zetasizer Nano ZS (Malvern Ins. Ltd., London, UK) was used to find the ζ-potential of sEVs. The values of the ζ-potential (three measurements) were averaged.

### 4.7. Confocal Microscopy

MSCs were cultured on sterile cover glasses overnight, stained with Hoechst 33342 for 1 h, then with a lipid dye, 3,5-bis-[(E)-2-phenylethenyl]-BDP (BDP) (Lumiprobe, Moscow, Russia), for 5 min, fixed with PFA, polymerized with Mowiol 4.88 medium (Calbiochem, Darmstadt, Germany), and analyzed. Prior to confocal microscopy, sEVs (10 µL) were stained with BDP for 5 min and washed twice in PBS. The sediment of sEVs was deposited on a dry cover glass, dried, washed with DPBS, and polymerized with Mowiol. The slides were analyzed using an Eclipse TE2000 confocal microscope (Nikon, Tokyo, Japan).

### 4.8. Multiplex Analysis of Cytokines

The quantity of cytokines/chemokines inside sEVs was determined by the standard 41-plex human cytokine/chemokine magnetic bead panel using a FLEXMAP 3D cytometer (EMD Milipore, Billerica, MA, USA). To this end, three 25 µL aliquots of sEVs were analyzed according to the manufacturer’s instructions. The data were analyzed automatically by the xPONENT software, version 3.1 (EMD Millipore).

### 4.9. Animals

To reproduce a model of hereditary AD, transgenic 5xFAD (TG6799) mice were used, which are characterized by accelerated deposition of amyloid plaques in the brain and the development of spatial memory loss [73]. These mice co-express Swedish (K670N/M671L), Florida (I716V), and London (V717I) mutations in human amyloid precursor protein (APP) 695, and M146L and L286V mutations in presenilin I, with both transgenes expressed under the Thy1 promoter. Transgenic animals were obtained from Charles River Laboratory (Wilmington, MA, USA). Mice were kept in a specially equipped room at a temperature of 21–23 °C with free access to water and standardized food. All experiments were performed in accordance with the Guidelines for the Care of Laboratory Animals (NIH) and the rules of laboratory practice (Order of the Ministry of Health of the Russian Federation dated 1 April 2016, No. 199n). The experimental protocol was validated by the Commission on Biosafety and Bioethics of ICB RAS (Approval ID: 5/062020, date: 12 June 2020). The studies were carried out on F1 offspring from crossing transgene carriers with SJL/C57BL6 mice (IBCh Affiliation, Pushchino, Moscow district, Russia). Genotyping of parental pairs and newborn 5xFAD mice was performed using PCR DNA analysis with the primers 5′-AGGACTGACCACTCGACCAG-3′ and 5′-CGGGGGTCTAGTTCTGCAT-3′, kits for DNA extraction, and PCR according to the manufacturer’s method (BioLabMix, Novosibirsk, Russia).

### 4.10. Primary Culture of 5xFAD Mouse Hippocampi

To minimize animal suffering and preserve the lives of parturient female mice, we refrained from utilizing strip operations and invasive surgical interventions. Instead, we adopted a methodology centered on natural childbirth without surgical manipulations. This approach permitted the avoidance of additional stress and trauma to the organism, thereby preserving physiological integrity and viability.

Subsequent work involved the acquisition of primary cell cultures from the hippocampi of newborn (0–1 day old) animals. For this purpose, we employed a rapid guillotining method under laboratory conditions. This technique ensures a swift and humane termination of the animals, thereby minimizing the risk of inflicting additional suffering.

In this study, we utilized animals from different litters to mitigate the effects of intra-litter variation on the experimental outcomes. Furthermore, to enhance the heterogeneity of the studied cultures, the hippocampi of the animals were pooled and homogenized in groups of four to five prior to mechanical processing. This methodology allowed for a diversified sample and reduced the potential influence of genetic and environmental factors associated with the origins of animals from the same litter.

Mechanically minced hippocampi were then treated with Trypsin/EDTA solution (Gibco, New York, NY, USA). The cell suspension obtained by enzymatic and mechanical dissociation was added to the wells of a 12-well plate coated with Poly-D-Lysine (Gibco, NY, USA) and 1 mL of Neurobasal Medium (Gibco, USA) containing 2% bioactive B-27 Supplement (Gibco, USA), and 1% Penicillin–Streptomycin–Glutamine (Gibco, USA). The medium was changed every three days. The culture plates were kept in a CO_2_ incubator at 37 °C and 5% CO_2_. All experiments were performed on the 14th day of cultivation. For confocal microscopy, the cultures were prepared in a similar manner on sterile cover glasses placed in 6-well plates. Each study was conducted in three biological replicates and three independent experiments were conducted.

### 4.11. Staining of sEVs from MSCs

To analyze the localization of sEVs in the primary culture of mouse hippocampi, sEVs were preliminarily stained with DiI (1,1′-Dioctadecyl-3,3,3′,3′-Tetramethylindocarbocyanine perchlorate, ThermoScientific, Waltham, MA, USA), which has an absorption maximum of 549 nm and 565 nm radiation. In short, 1 mL of sEVs with a concentration of 5 × 10^10^ particles/mL, according to NTA, was incubated with 10 µL of 20 mM DiI dissolved in dimethyl sulfoxide for 20 min, after which sEVs were washed three times in saline. Staining of sEVs and the sedimentation resulted in some particle aggregation, however, analysis by particle numbers showed a small aggregated fraction 200–250 nm in diameter (Appendix A). The resulting precipitate was dissolved in saline and 80 × 10^6^ sEVs added to cell cultures. Cells were fixed after 3 h, and then confocal microscopic analysis of the labeled sEV localization in the cells was performed.

### 4.12. Cell Viability Analysis

The influence of sEVs on the viability of hippocampal Tg cultures was assessed 24 h later. Cell viability in the hippocampal cultures was analyzed by staining with Hoechst 33,342 and propidium iodide (PI) (ThermoScientific, Waltham, MA, USA) for 30 min. The number of positive cells was analyzed by the ImageJ program. Statistical processing of the results was carried out using one-factor analysis ANOVA followed by post-hoc analysis and application of Tukey’s HSD test.

### 4.13. Effect of sEVs on Synaptogenesis and Immunoreactivity to Aβ

Hippocampal cultures were cultivated with sEVs (4 × 10^6^, 40 × 10^6^, 80 × 10^6^, or 160 × 10^6^ per culture) for 24 h, fixed for 10 min with 4% paraformaldehyde, perforated with 0.1% Triton X-100 solution, and incubated for 1 h with 2% bovine serum albumin in saline to block non-specific binding. To analyze synaptogenesis, cells were stained with primary guinea pig antibodies to synaptophysin-1 (SYSY Synaptic system, RRID:AB_1210382, Goettingen, Germany, dilution 1:1000) overnight at 4 °C, followed by secondary anti-guinea pig-Alexa Fluor 405 (Abcam, Cambridge, UK, RRID:AB_2827755, 1:1000) antibody. Image processing was performed using the ImageJ software 1.8.0_112 32 /64 bit. The statistical analysis of the results was carried out using one-factor variance analysis using the Student–Newman–Keuls criteria. To estimate the effect of sEVs on neuronal density, the cells were stained with primary antibodies to the neuronal marker MAP2 (Abcam, RRID:AB_776174, 1:100) and the corresponding secondary antibodies Alexa Fluor 594 (Abcam, RRID:AB_2782993, 1:200). The effect of sEVs on Aβ in Tg culture was assessed using anti-beta-amyloid 1–42 antibody (Abcam, RRID:AB_296881, 1:200) and the secondary antibodies Alexa Fluor 594 (Abcam, RRID:AB_2650602, 1:1000). Quantification of Aβ-positive areas was analyzed using the ImageJ software. Statistical processing of the results was carried out using univariate analysis of variance using the Student–Newman–Keuls criteria. Immunoreactivity to astrocytes was analyzed using primary antibodies to GFAP (Abcam, RRID:AB_305808, 1:500) and the secondary antibodies Alexa Fluor 488 (Abcam, RRID:AB_2630356, 1:2000). Nuclei were visualized by Hoechst 33,342. Microscopic analysis was performed using an Axio Imager.Z1 microscope (Carl Zeiss, Oberkochen, Germany) and Leica TCS SP5 confocal inverted microscope (Leica Microsystems GmbH, Vienna, Austria). An automated JuLI Stage system was also used for imaging. Data obtained from each well of the culture plate were randomly selected using a random number generator. This approach minimized possible systematic bias associated with the subjective choice of the researcher. Additional blinding was implemented at the stage of counting and analyzing the results; the researcher who performed these procedures did not know to which experimental group the evaluated data belonged. The in vitro experiments were conducted by the Pushchino group (DYZ conducted and analyzed them), and then the blinded images were sent to IBCh (EVS) and we compared the results.

### 4.14. Statistics

Statistical analysis of the results was performed using the Sigma Plot 12.5 software. Graphs were plotted using Prism 10.4.0 (Boston, MA, USA). Data are presented in the form of mean values and the standard error of the mean. The hypothesis of normal distribution was tested using the Shapiro–Wilk test. One-factor ANOVA of variance with a post-hoc analysis and comparison of groups by Tukey’s method (in the cases of cell viability and synaptophysin data analysis) and Student–Newman–Keuls (in the case of data analysis of the immunopositivity area to markers of Aβ) were used to compare differences between groups and to determine statistical significance. Significance levels of *p* < 0.05 were considered statistically significant. Each study was conducted in three biological replicates.

## 5. Conclusions

To date, the exact mechanisms of the neuroprotective and neuroregenerative effects of MSC sEVs remain unknown, which is a barrier to executing clinical trials. The positive effect of sEVs for treatment can depend on their number and source, protein composition, amino acid, lipid, mRNA, microRNA, and mtRNA content, and other biologically active components. Currently, the effects of sEVs are studied in different models to develop perspective therapeutics for AD. Our results support earlier findings of sEV effects in a new AD model and sEV isolation method. It was shown that sEV cargo can play a role in the protective activity. In general, the obtained results indicate that sEVs are currently one of the most promising agents for AD treatment.

## Figures and Tables

**Figure 1 ijms-26-04026-f001:**
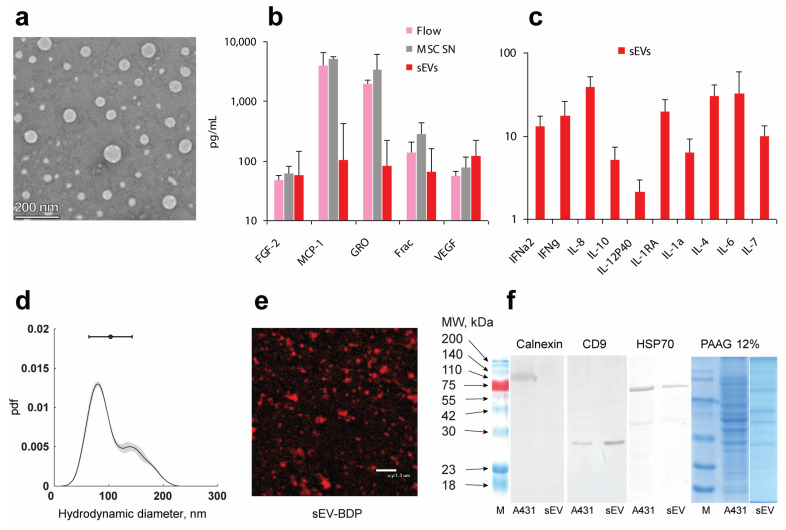
Characteristics of sEVs purified by asymmetric depth filtration. (**a**): Heterogeneity of sEVs as determined by TEM. (**a**–**c**): Protein content of MSC supernatant (MSC SN), and the flow through after sEV purification (Flow) and in sEVs found by 41-plex assay, respectively. Frac denotes fractalkine. A total of 6 flow-through, 12 MSC SN, and 6 sEV samples were used (see Appendix A for details). (**d**,**e**): NTA analysis of the hydrodynamic diameter of sEVs (pdf—probability density function) (**d**) and confocal image (**e**) of sEVs stained with lipid dye BDP (red). (**f**): Western blot and 12% PAAG of sEVs and control cells (A431) stained with antibodies to calnexin, CD9, and HSP70 (M—protein ladder).

**Figure 2 ijms-26-04026-f002:**
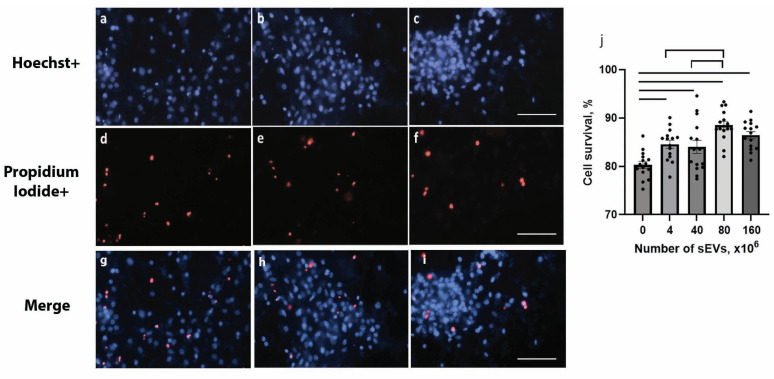
The effect of sEVs on cell survival in the primary culture from the hippocampi of Tg 5xFAD mice. Representative microphotographs of primary neuronal culture of transgenic 5xFAD mice incubated without (**a**,**g**) or with 40 (**b**,**h**) or 80 (**c**,**i**) ×10^6^ sEVs. (**a**–**c**): Cell nuclei stained with Hoechst 33,342 (blue). (**d**–**f**): Dead cells stained with propidium iodide (red). (**g**–**i**): Merged blue and red channels. Scale bar = 25 µm. (**j**): Cell survival. Significant differences between the untreated control and after sEV treatment are shown with bars; significant differences between the sEV doses are shown with brackets (*p* < 0.05). F = 10.692, *p* < 0.001. Cell survival (%) (mean ± SEM): control (80.32 ± 0.72), 4 × 10^6^ (84.57 ± 0.84), 40 × 10^6^ (84.01 ± 1.36), 80 × 10^6^ (88.48 ± 0.83), and 160 × 10^6^ (86.43 ± 0.75) sEVs.

**Figure 3 ijms-26-04026-f003:**
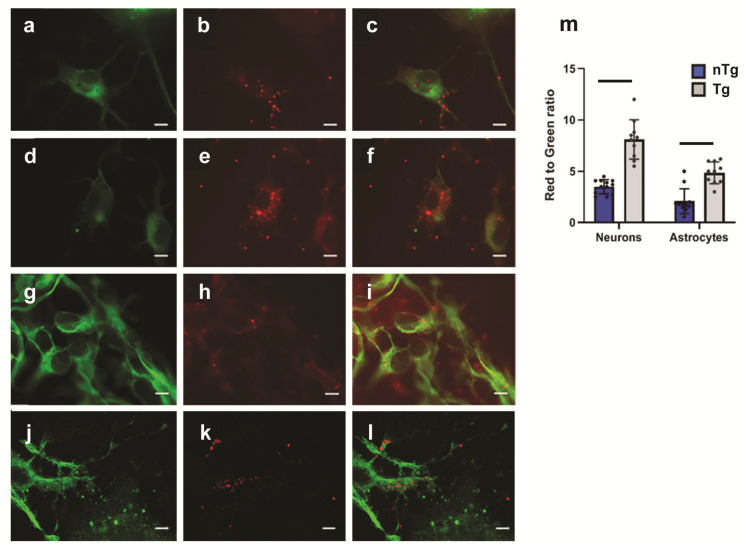
Colocalization of sEVs with neurons and astrocytes in nTg and Tg cultures. Single and merged staining of sEVs (red) with MAP2-positive neurons (green) in nTg (**a**–**c**) and Tg (**d**–**f**) cultures. (**g**–**l**): Single and merge staining of sEVs (red) with GFAP-positive astrocytes (green) in nTg (**g**–**i**) and Tg (**j**–**l**) cultures. Three independent experiments were conducted showing reproducible results, and representative images are provided. Scale bars = 5 µm. (**m**): Average fluorescence of sEVs was normalized to green fluorescence and averaged. Bars signify statistical differences (*p* < 0.05).

**Figure 4 ijms-26-04026-f004:**
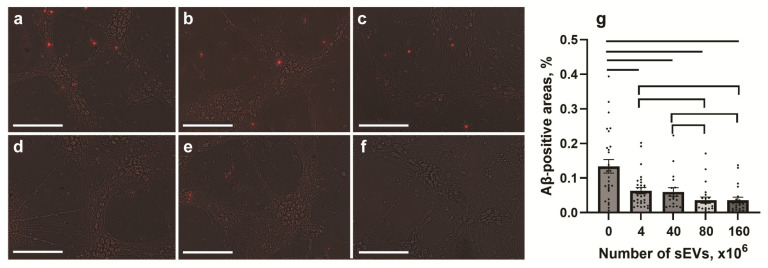
Dose-dependent response to sEVs treatment in the expression of Aβ by Tg cells. Representative microphotographs of Aβ-immunopositivity (red) in Tg cells incubated for 24 h with 0 (**a**), 4 × 10^6^ (**b**), 40 × 10^6^ (**c**), 80 × 10^6^ (**d**), and 160 × 10^6^ (**e**) sEVs, and in nTg (**f**) cells. Scale bars = 250 µm. (**g**): Aβ-positive areas, %. Statistically significant differences between the untreated and sEV-treated Tg are shown with bars; significant differences between the sEV doses are shown with brackets (*p* < 0.05). Aβ positive area, % (mean ± SEM): control (0.134 ± 0.020), 4 × 10^6^ (0.063 ± 0.010), 40 × 10^6^ (0.060 ± 0.012), 80 × 10^6^ (0.036 ± 0.009), and 160 × 10^6^ (0.037 ± 0.008) sEVs.

**Figure 5 ijms-26-04026-f005:**
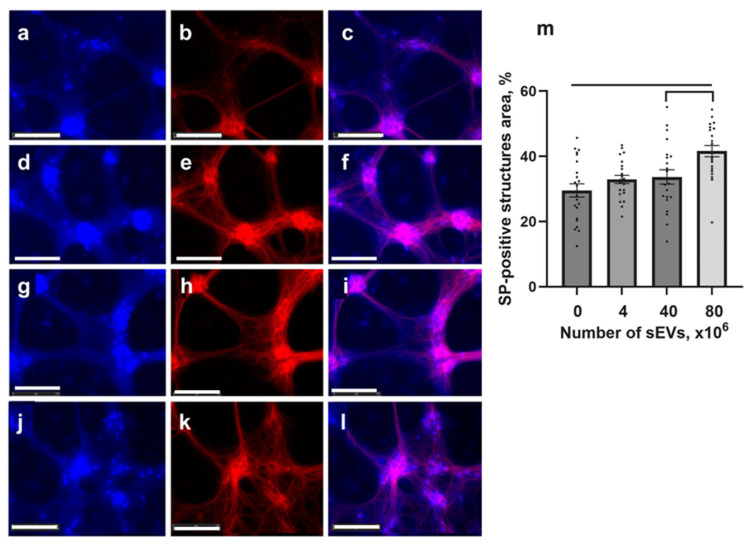
Immunopositivity of synaptophysin (blue) and neurons (MAP2, red) in Tg cultures. Separated and merged channels after staining of synaptophysin (SP, blue) and MAP2 (red) before (**a**–**c**) and after incubation with 4 × 10^6^ (**d**–**f**), 40 × 10^6^ (**g**–**i**), and 80 × 10^6^ (**j**–**l**) sEVs. Scale bars = 125 µm. (**m**) Changes in the area of SP-positive structures in Tg cultures after introduction of sEVs. Statistically significant differences between the untreated Tg control and treated Tg cultures using different doses of sEVs are shown with bars; significant differences between the sEV doses are shown with brackets (*p* < 0.05). F = 10.830, *p* < 0.001. Positive area, % (mean ± SEM): control (29.55 ± 2.04), 4 × 10^6^ (32.92 ± 1.27), 40 × 10^6^ (33.67 ± 2.20), and 80 × 10^6^ (41.62 ± 1.73) sEVs.

## Data Availability

All data needed to evaluate the conclusions in this paper are present in the main text and Appendix A. The datasets used and/or analyzed during the current study are available from the corresponding author on reasonable request.

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
