# Peer review of "Effect of Small Extracellular Vesicles Produced by Mesenchymal Stem Cells on 5xFAD Mice Hippocampal Cultures"

_ijms, 2025, doi:10.3390/ijms26094026_

Round 1

Reviewer 1 Report

Comments and Suggestions for Authors

The authors were using MSC derived EVs to treat 5xFAD mice hippocampus culture. The results of reduction of Aβ and increased the synaptic density were shown to improve brain cell function. But there is still something need to be improved on the experimental setting.

  1. More characterization needed for MSC derived sEV. Soluble proteins tested by the cytokine panel did not show any enrichment of any cytokines (Fig1c, S1), and the author did not prove the link of any cytokines with the function. If it’s not because of these cytokines, there is no reason to show these data. And if it’s not enriched in sEV, why did not just use the MSC medium? Also, the medium will be a good control for the rest of function assays.
  2. In Fig1f-g, lipid staining does not mean anything. Same as cytokines, if it’s not linked with function, there will be no need to present. Some lipophilic dyes also form aggregates that will look similar to EVs in many assays like flow cytometry and microscopy.
  3. Primary culture of hippocamps needs to be well characterized to show the ratio of different cell types/viability. Is the MSC EV more enriched in Neurons?
  4. For Aβ staining, area and puncta dots intensity should both be quantified. And colocalization of fluorescent labeled EVs with Aβ puncta will give a better understanding to interpret the result.

Minor:

Page 4, line 110: Title

Page9, line 286: use * or X instead of @

Comments on the Quality of English Language

Good.

Author Response

More characterization needed for MSC derived sEV. Soluble proteins tested by the cytokine panel did not show any enrichment of any cytokines (Fig1c, S1), and the author did not prove the link of any cytokines with the function. If it’s not because of these cytokines, there is no reason to show these data. And if it’s not enriched in sEV, why did not just use the MSC medium? Also, the medium will be a good control for the rest of function assays.

We have characterized sEV according to the position statement from the International Society for Extracellular Vesicles [23]. As far as cytokines, some were found and we decided to include results in Fig.1. The reason is that the mechanisms of sEV activity are still not well understood. It can be chemokines, cytokines, other components in sEVs involved, modernly it is not known which factors mediate protective effects of sEVs. The aim of this paper was to analyze whether there is any effect of sEVs on hippocampal cells. Which component of sEVs is responsible, it is not easy to say. We cannot say with confidence that cytokines play a role in the effect.

The data on humoral factor sEV content have independent values, we have shown the correlation, it has sense as to why sEVs should contain something different from what was in the cells.

As to why not to use culture medium in the cell experiments (if we understand the comment correctly), culture medium has many other components which will not pass BBB and in the end it may be difficult to make conclusions. The idea of our work is to use only sEVs in vivo as they are able to pass BBB via intranasal instillation excluding any other components of the culture medium that can be generated by cells into extracellular space.

In Fig1f-g, lipid staining does not mean anything. Same as cytokines, if it’s not linked with function, there will be no need to present..

We were to an extent interested to see stained MSCs but according to comments we have removed this image. About the cytokine function please see the previous answer. Moreover, to delivery of cytokines in the brain can stimulate inflammation, which can negatively affect the brain.   

The title of our article itself indicates that the subject of our study is to see the effect of sEVs isolated from Wharton's jelly MSCs of the umbilical cord. So we have to be sure that we used sEVs but not other cell organoids or dye conglomerates. According to the statement from the International Society for Extracellular Vesicles (Welsh, J.A.; Goberdhan, D.C.I.; O’Driscoll, L.; Buzas, E.I.; Blenkiron, C.; Bussolati, B.; Cai, H.; Di Vizio, D.; Driedonks, T.A.P.; Erdbrügger, U.; et al. Minimal Information for Studies of Extracellular Vesicles (MISEV2023): From Basic to Advanced Approaches. Journal of Extracellular Vesicles 2024, 13, e12404, doi:https://doi.org/10.1002/jev2.12404) sEVs must be positive for membrane bound proteins such as CD9, CD63, CD81, negative for the proteins associated to other intracellular compartments than sEVs such as calnexin, on the other hand. sEV cargo should largely reflect the content of the cells which produce them. That is why the significant part of our work was devoted to the characterization of sEVs isolated from MSCs, including intracellular proteins such as heat shock proteins, and presence of lipid layer. In this context, we were not necessarily interested in the functional significance of individual components (cytokines, lipids, and even HSP-70), the fact of their presence in sEVs was important, which we reflected in Fig. 1 (c-e) and in Table S1. An important result, proving that we are indeed dealing with sEVs, was Pearson’s correlation analysis which demonstrated that the concentration of the soluble factors in the flow-through supernatants after sEV purification and MSC before the purification correlated significantly (r=0.992, p<0.0001) (Table S1). The same correlation analysis demonstrated that sEVs protein content had some correlation with MSC SN data (r=0.504, p<0.005). These results demonstrated that sEVs cargo reflects the content of MSCs which produce them.  We did not use the medium either as an active compound or as a control for the effects of sEVs, although the medium was more enriched with some cytokines and lipid components. However the ratio of different components in the medium and in sEVs was different (Fig. 1 e-f and Table S1).

Some lipophilic dyes also form aggregates that will look similar to EVs in many assays like flow cytometry and microscopy

We agree with reviewer and it is important raising this concern. We were fully aware of this potential issue. To minimize such possibilities, we implemented several modifications to our protocol: optimized the staining procedure and most importantly, enhanced the washing steps to thoroughly remove unbound dye. Since this is an optimized procedure that we used previously we already know by previously having control sample (data not shown) that all unbound dye is removed. Hence we decided not to include nearly an empty image containing no labeled sEVs.

Primary culture of hippocamps needs to be well characterized to show the ratio of different cell types/viability.

We have previously shown that the transgenic culture is characterized by a higher density of astrocytes and a lower density of neurons compared to the non-transgenic culture, and also that a 14-17-day-old primary hippocampal cell culture from 5XFAD mice can be considered a valid cellular model of Alzheimer's disease (AD) and has been sufficiently previously studied by us (A. V. Chaplygina; D. Yu. Zhdanova; R. A. Poltavtseva; N. V. Bobkova Effect of Conversion Cocktail on Astrocyte and Neuronal Status in the Primary Hippocampal Culture of 5xFAD Mice with Angiotensin-Converting Enzyme 2 Inhibition. // Journal of Evolutionary Biochemistry and Physiology, 2024. vol. 60, 6. 2535–2546; Chaplygina, A. V., Zhdanova, D. Y. (2024). Effects of mitochondrial fusion and fission regulation on mouse hippocampal primary cultures: relevance to Alzheimer's disease. Aging Pathobiology and Therapeutics, 08-17; Chaplygina, A. V., Zhdanova, D. Y., Kovalev, V. I., Poltavtseva, R. A., & Bobkova, N. V. (2023). Interaction of MSCs with 5XFAD Mouse Hippocampal Cells in Primary Culture Depending on Cocultivation Method. Biochemistry (Moscow), Supplement Series A: Membrane and Cell Biology, 17(2), 156-168). In our study, we observed the presence of immunoreactivity to beta-amyloid in the cultured cells. It is important to note that the antibodies we used, Anti-beta Amyloid 1-42 antibody (Abcam, ab10148), interact with Aβ42 but do not show cross-reactivity with Aβ1-40, APP, sAPPβ, or sAPPα.

The area of ​​immunopositive staining of the astrocyte marker (GFAP – green) and the neuronal marker (MAP2 – red) in primary cell cultures of the hippocampus of non-transgenic (nTg) and transgenic (Tg) mice. nTg – Native primary culture of the hippocampus of the nTg mouse. Tg – Native primary culture of the hippocampus of the Tg 5xFAD mouse. Graphs show the density of neurons and astrocytes (in %).

** p < 0.05, one-way ANOVA with subsequent Dunn's post-tests. Scale bar – 250 µm.

Is the MSC EV more enriched in Neurons?

sEVs co-localize to a greater extent with neurons, which is especially noticeable in the transgenic culture. Targeted delivery to these cells could potentially enable partial or complete restoration of the functional state of affected neurons. It can be assumed that such selectivity is partially responsible for the normalization of neuronal activity and increased synaptogenesis observed under the influence of exosomes.

Co-localization of sEVs with neurons and astrocytes in nTg and Tg cultures. Single and merged staining of sEVs (red) with MAP2 positive neurons (green) in nTg (a-c) and Tg (d-f) cultures. g-l: Single and merge staining of sEVs (red) with GFAP positive astrocytes (green) in nTg (g-i) and Tg (j-l) cultures. Three independent experiments were conducted with close results and the repre-sentative images are shown. Scale bars = 5 µm. m: Average fluorescence of sEVs was normalized to green fluorescence and averaged. Bars signify statistical difference (p<0.05).

For Aβ staining, area and puncta dots intensity should both be quantified. And colocalization of fluorescent labeled EVs with Aβ puncta will give a better understanding to interpret the result.

We appreciate the reviewer’s suggestion to quantify puncta intensity. However, in our imaging setup, fluorescence intensity of individual puncta cannot be directly measured due to limitations in our acquisition and analysis methods. Although the puncta are visible, their small size, combined with our technical limitations, makes intensity measurements less reliable. Instead, we quantified the total stained area, which provides a more reproducible and biologically relevant measure of Aβ accumulation in our system. 

Representative microphotograph of Aβ (red) on Tg cells culture.

In this case, the use of area for calculating possible changes is much more important than calculation by the number of cells, since the area occupied by beta-amyloid reflects the pathological process to a greater extent. Aβ Amyloid 1-42 antibody (Abcam, ab 10148) was used. This antibody is reactive with Aβ 42 and does not cross-react with Aβ Amyloid 1-40, full length APP, sAPP β or sAPP α. In addition, the results obtained using enzyme-linked immunosorbent assay (ELISA) and the AssayKit for Amyloid Beta Peptide1-42 (Ab1-42) reagent indicate the content of Aβ (1.37±0.09 pg/ml) in cell cultures at 14 days, which is confirmed in experiments with immunocytochemical staining for Aβ.

And colocalization of fluorescent labeled EVs with Aβ puncta will give a better understanding to interpret the result.

The colocalization of fluorescently labeled sEVs with Aβ in a 5xFAD mouse hippocampal cell culture may not provide a comprehensive understanding of the results for several reasons:

  • Colocalization only indicates spatial proximity between EVs and Aβ but does not provide information about the functional interaction or biological consequences of this proximity. It does not reveal whether EVs influence Aβ aggregation, clearance, or toxicity.
  • Colocalization could not answer on question whether EVs are actively involved in Aβ clearance or passively associated with Aβ deposits.
  • Colocalization alone does not provide mechanistic insights into how EVs might modulate Aβ aggregation, degradation, or cellular uptake. Additional experiments are required to elucidate these mechanisms.

Minor:

Page 4, line 110: Title

Page9, line 286: use * or X instead of @

Thank you for pointing out the typos. We corrected them as suggested. Please let us know if we missed something.

Reviewer 2 Report

Comments and Suggestions for Authors

In this manuscript, Zhdanova et al investigate the effect of Wharton’s jelly MSC-derived sEVs on hippocampal cells from 5xFAD transgenic mice with Alzheimer’s disease mutations. They found that the MSC-sEVs are non-toxic, with a stepped dose based response, colocalize with neurons and astrocytes and can reduce the amyloid beta plaque areas in the transgenic mice. This work is organized and provides a sequential assessment of the MSC-sEVs. Below I suggest a few comments to strengthen the manuscript.

Reviewer Comments:

(General Comments):

  • The sEVs showed two distinct size populations in the NTA which the authors acknowledged in text. Why was there this distribution of sEVs and what impact could this have? Could only 1 of the two sEV populations be resulting in the effects observed? Please discuss.
  • In figure 2, why was Hoechst-33342 used as the ‘live-cell’ marker? Hoechst-33342 can stain dead nuclei too. Please update with another live-cell marker (e.g. Calcein-AM)
  • Also, for Figure 2, include the merged (overlayed) image as well, so it’s easier to see the ratio of live vs dead cells in a give area
  • In Section 2.2, please explain why the specific does of 4, 40, 80, 160 were used, since they are not typically concentrations, what was the rationale?
  • In Figure 3, show the control as well (dye-only) to confirm that the red speckles seen are in fact sEVs
  • Also, for Figure 3 provide a z-stack of the images, so we can confirm the sEVs aren’t just settled onto the surface of the neurons/astrocytes.
  • To further bolster the co-localization claims, the authors should quantify using flow cytometry or, if available, using imaging flow cytometry.
  • Why were MSC-sEVs specifically selected for this study? Please explain/include this rationale to strengthen the study.
  • Please include the MSC characterization (surface marker assessment) into supplementary to confirm the cell population.

(Minor Comments):

  • Add labels to the imaging figure panels so it’s easier to follow instead of reading the full caption (e.g. in Figure 2A:, label the top row Hoechst+, and the bottom row Propidium Iodide+ in the figure itself, so it’s quicker for the reader to follow, similar to how Figure 1f and g have labels beneath them).
  • In section 2.2, the authors states that there ‘was a dose-dependent’ response, but the 4 and 40x10^6 show very similar values. It looks almost like a stepped response, saturating at 80x10^6. Why do the authors think this is the case?
  • For relevant plots, please include sample sizes (e.g. Fig 1C, Fig 2g etc.)

Author Response

First of all, we would like to express our deep gratitude to you for your thoughtful work on our paper and a number of valuable comments on the presented results and their discussion. We hope that the article has become much better taking into account your comments. Below, we have tried to answer the questions you asked.

In this manuscript, Zhdanova et al investigate the effect of Wharton’s jelly MSC-derived sEVs on hippocampal cells from 5xFAD transgenic mice with Alzheimer’s disease mutations. They found that the MSC-sEVs are non-toxic, with a stepped dose based response, colocalize with neurons and astrocytes and can reduce the amyloid beta plaque areas in the transgenic mice. This work is organized and provides a sequential assessment of the MSC-sEVs. Below I suggest a few comments to strengthen the manuscript.

The sEVs showed two distinct size populations in the NTA which the authors acknowledged in text. Why was there this distribution of sEVs and what impact could this have? Could only 1 of the two sEV populations be resulting in the effects observed? Please discuss.

Thank you for pointing this out. The isolation method does result in a rather heterogeneous population of sEVs. It is possible that these populations if fractionated by size can differ and have different effect on cells.

It is important that both detected particle populations fall well within the conventionally defined hydrodynamic size range for sEVs (50–200 nm), according to the classification presented in current literature (Pegtel, D. M., & Gould, S. J. (2019). Exosomes. Annual review of biochemistry, 88, 487-514.; Zhou, Y., Seo, J., Tu, S., Nanmo, A., Kageyama, T., & Fukuda, J. (2024). Exosomes for hair growth and regeneration. Journal of bioscience and bioengineering, 137(1), 1-8). The size heterogeneity within the sEV range may reflect natural variation, and attempts to separate these naturally co-secreted sEVs subpopulations could create artificial biological contexts and potentially overlook synergistic effects. Since both populations meet the sEV criteria for sEVs, we propose that their combined use better reflects the natural biological activity of MSC-derived EVs.

A close examination of the size distribution of sEVs shows that one population accounts for approximately 75% of all sEVs with an average hydrodynamic diameter of about 100 nm. This population may have the greatest contribution to the observed effects but this is a speculation at this point and we cannot exclude that a small fraction of sEVs may have a greater effect than other fractions.

However, we fully agree that more sophisticated analyses of individual vesicle populations may provide deeper insights in future studies. For example, we can use the asymmetric flow field flow fractionation (AFFF) technique to obtain sEVs with a narrow size distribution (see Zhang, H., Lyden, D. Asymmetric-flow field-flow fractionation technology for exomere and small extracellular vesicle separation and characterization. Nat Protoc 14, 1027–1053 (2019). https://doi.org/10.1038/s41596-019-0126-x). We are planning in the near future to perform exactly that to compare and explore more deeply into what subpopulation of sEVs with what cargo is responsible for the reported effect.

In figure 2, why was Hoechst-33342 used as the ‘live-cell’ marker? Hoechst-33342 can stain dead nuclei too. Please update with another live-cell marker (e.g. Calcein-AM)

The determination of cell viability using dual staining with Hoechst 33342 and propidium iodide (PI) is a reliable, simple, and versatile method widely recognized and applied in scientific research (Chan, L. L., Wilkinson, A. R., Paradis, B. D., & Lai, N. (2012). Rapid image-based cytometry for comparison of fluorescent viability staining methods. Journal of fluorescence, 22(5), 1301-1311).

Its key advantages include high accuracy in differentiating live and dead cells, ease of execution, the ability to visualize cell morphology, and compatibility with other methods. Due to its cost-effectiveness, minimal impact on cells, ability to maintain cell viability during staining, and potential for dynamic studies—such as tracking changes in cell viability in real time—this method remains one of the most popular approaches for assessing cell viability under various experimental conditions.

In Figure 2 we demonstrated that cell nuclei were stained with Hoechst 33342 but did not specify that Hoechst 33342 was used solely as a "live-cell" marker. Indeed, Hoechst 33342 can stain both dead nuclei and live cell nuclei. Therefore, it was used in combination with propidium iodide, a membrane- dye that penetrates only cells with damaged membranes. This dual staining approach allows for precise discrimination between live and dead cells, enhancing the accuracy of cell viability assessment.

Also, for Figure 2, include the merged (overlayed) image as well, so it’s easier to see the ratio of live vs dead cells in a give area

Thank you for the suggestion. We included the merged figures.

Figure. The effect of the sEVs on cell survival in the primary culture from the hippocampi of Tg 5xFAD mice. Representative microphotographs of primary neuronal culture of transgenic 5xFAD mice incubated without (a, d) or with 40 (b, e) or 80 (c, f) x106 of sEVs. a-c: Cell nuclei stained with Hoechst 33342 (blue); d-f Dead cells stained with propidium iodide (red). The scale is 25 µm. g: Cell survival.  Significant difference from the control is shown with the bars, between sEV doses – by the brackets (p<0.05).

In Section 2.2, please explain why the specific does of 4, 40, 80, 160 were used, since they are not typically concentrations, what was the rationale?

In this study, sEV concentrations of 4, 40, 80, and 160×10⁶ particles were used for administration into hippocampal cell cultures from 5xFAD mice to investigate the dose-dependent effects of these vesicles on cellular processes associated with neurodegeneration and cognitive function. Earlier In behavioral tests, we have shown that intranasal administration of sEVs at a concentration of 80×10⁶ particles demonstrated a positive effect on memory improvement in a mouse model of sporadic AD (Zhdanova, D.Yu.; Poltavtseva, R.A.; Svirshchevskaya, E.V.; Bobkova, N.V. Effect of Intranasal Administration of Multipotent Mesenchymal Stromal Cell Exosomes on Memory of Mice in Alzheimer’s Disease Model. Bull Exp Biol Med 2021, 170, 575–582, doi:10.1007/s10517-021-05109-3). This suggests that this concentration is biologically active and capable of modulating neuronal functions in vivo. The use of lower (4 and 40×10⁶ particles) and higher (160×10⁶ particles) concentrations allows for the assessment of how cellular responses vary depending on the dose. This is important for determining the optimal concentration that may exert maximal therapeutic effects without adverse side effects, as well as for understanding the mechanisms of sEV action at the cellular level. Lower concentrations may help identify threshold values required to trigger biological effects, while higher concentrations allow evaluation of whether the positive effect persists with increased dosage or whether potential toxic effects emerge. Thus, this approach provides a comprehensive understanding of the dose-dependent influence of extracellular vesicles on hippocampal cell culture, which is a crucial step in further exploring their therapeutic potential in the context of neurodegenerative diseases.

In Figure 3, show the control as well (dye-only) to confirm that the red speckles seen are in fact sEVs

Centrifugation was fulfilled to remove the unbound dye. Primarily sEV were isolated by soft centrifugation. After dye staining we have measured sEV diameters (Fig.1). sEV slightly enlarged and a small fraction aggregated (arrow). You will find this figure is in the supplemental information.

Also, for Figure 3 provide a z-stack of the images, so we can confirm the sEVs aren’t just settled onto the surface of the neurons/astrocytes.

Thank you for the suggestion. We included 2 gif files which show z-stack of images. This shows that they are not simply settled on the surface.

To further bolster the co-localization claims, the authors should quantify using flow cytometry or, if available, using imaging flow cytometry.

Indeed, these procedures would have provided a greater amount of data for analysis; however, they were not conducted because we believe the method we employed is sufficient for the objectives of the present study.

Why were MSC-sEVs specifically selected for this study? Please explain/include this rationale to strengthen the study.

 It should be noted that this work was a logical continuation of a whole cycle of our studies devoted to studying the effectiveness of intracerebral and intravenous transplantation of different types of cells on the course of the neurodegenerative process in a model of sporadic Alzheimer's disease (Poltavtseva, R. A., Samokhin, A. N., Bobkova, N. V., Alexandrova, M. A., & Sukhikh, G. T. (2020). Effect of Transplantation of Neural Stem and Progenitor Cells on Memory in Animals with Alzheimer's Type Neurodegeneration. Bulletin of experimental biology and medicine, 168(4), 589–596. https://doi.org/10.1007/s10517-020-04758-04; Bobkova, N. V., Poltavtseva, R. A., Samokhin, A. N., & Sukhikh, G. T. (2013). Therapeutic effect of mesenchymal multipotent stromal cells on memory in animals with Alzheimer-type neurodegeneration. Bulletin of experimental biology and medicine, 156(1), 119–121. https://doi.org/10.1007/s10517-013-2293-z; Panchenko, M. M., Poltavtseva, R. A., Bobkova, N. V., Vel'meshev, D. V., Nesterova, I. V., Samokhin, A. N., & Sukhikh, G. T. (2014). Localization and differentiation pattern of transplanted human multipotent mesenchymal stromal cells in the brain of bulbectomized mice. Bulletin of experimental biology and medicine, 158(1), 118–122. https://doi.org/10.1007/s10517-014-2706-7). The most striking effects were observed with MSCs. It is quite logical that we then investigated the behavioral effects of sEVs isolated from MSCs and the possibility of their intranasal administration (Zhdanova, D. Y., Poltavtseva, R. A., Svirshchevskaya, E. V., & Bobkova, N. V. (2021). Effect of Intranasal Administration of Multipotent Mesenchymal Stromal Cell Exosomes on Memory of Mice in Alzheimer's Disease Model. Bulletin of experimental biology and medicine, 170(4), 575–582. https://doi.org/10.1007/s10517-021-05109-3).

In addition to the above sEVs isolated from MSCs were chosen for this study due to several key advantages over sEVs derived from other cell types:

  • MSCs have been studied for their ability to modulate inflammation, promote tissue regeneration, and exert neuroprotective effects. These properties are largely mediated by their paracrine activity, including the release of sEVs. MSC-sEVs carry bioactive molecules (proteins, lipids, RNAs) capable of influencing processes relevant to neurodegeneration, such as synaptic plasticity, neuronal survival, and the reduction of neuroinflammation.
  • Unlike whole MSCs, sEVs lack proliferative capacity, eliminating risks associated with uncontrolled differentiation or tumor formation. MSC-sEVs exhibit low immunogenicity, reducing the likelihood of adverse immune responses upon administration—a critical feature for potential clinical translation. Furthermore, their high biocompatibility makes them a safe vehicle for therapeutic molecule delivery.
  • Due to their small size and lipid bilayer membrane, MSC-sEVs efficiently cross biological barriers, including the blood-brain barrier (BBB). This property renders them ideal candidates for delivering therapeutic agents to the central nervous system. Additionally, sEVs provide a standardized, cell-free therapeutic product with extended shelf life and simpler quality control compared to live-cell therapies.
  • In contrast to EVs isolated from other cell types (e.g., immune or tumor cells), MSC-sEVs possess a unique molecular profile that promotes tissue repair and inflammation modulation. Moreover, MSCs are easier to culture and scale up for large-scale EV production, which is essential for both research and clinical applications.
  • Thus, the selection of MSC-sEVs for this study is justified by their unique biological properties, which position them as effective tools for targeting AD-related pathological mechanisms, alongside their favorable translational prospects.

Please include the MSC characterization (surface marker assessment) into supplementary to confirm the cell population.

The MSCs were obtained from Wharton's jelly of the umbilical cord by employees of the Federal Institution, the Academician V.I. Kulakov Research Center for Obstetrics, Gynecology and Perinatology of the Ministry of Health of the Russian Federation who have a special permit for this type of work. The primary culture of MSCs was isolated from Wharton's jelly of the umbilical cord obtained after cesarean section from different (at least five) healthy examined women in labor. The material was collected with the written informed consent of the donors. The tissue samples were mechanically crushed and placed in a 0.1% solution of type I collagenase (Gibco, USA) for 60 min at 37°C. After incubation, the suspension was precipitated by centrifugation for 3 min at 200g. The pellet was resuspended in DMEM (Gibco, USA)-F12 (PanEco, Russia) (1:1) with the addition of 10% fetal bovine serum (Gibco, USA) and 1% Penicillin-Streptomycin-Glutamine (Gibco, USA) and placed in 25 cm2 culture flasks (Corning). The cells were cultured at 37°C and 5% CO2. Cells that reached 80% confluency were detached with 0.05% trypsin (PanEco, Russia).

The phenotype of the isolated MSCs was characterized by flow cytometry using specific markers. Fluorescence intensity was analyzed using a FACSCalibur flow cytometer and BD CellQuest Pro software (BD Biosciences, USA). The cell population was isolated by forward (FSC) and side (SSC) light scattering parameters. The expression of markers such as CD90, CD105, CD73, CD19, and HLA-DR was assessed. The expression level of the studied marker was assessed by the fluorescence intensity histogram. Immunocytochemical staining was performed using monoclonal antibodies conjugated with phycoerythrin. For analysis, the cells were removed from the surface of the flask with a trypsin solution and washed twice with staining buffer (PBS, 1.0% fetal bovine serum, and 0.1% sodium azide). 15 μl of labeled antibodies to one of the studied surface markers were added to the cells, bringing the volume of the cell suspension to 100 μl. The corresponding isotype antibodies were added to the samples serving as a negative control. The cell suspension was incubated at +4°C for 1 hour. After incubation, the cells were washed 2 times in 1 ml of staining buffer and fixed in 0.5 ml of 2% paraformaldehyde solution. The resulting cell suspension was filtered through a filter with a pore diameter of 30 μm to exclude cell aggregates. Fluorescence intensity was analyzed using a FACSAria flow cytofluorimeter-sorter (Becton Dickinson Co.). In further experiments, only MSCs that had undergone no more than 6 passages were used.

As a result of the cytometric analysis, it was shown that the obtained culture of MSCs isolated from Wharton's jelly of human umbilical cord expressed markers CD73, CD90, CD105, recommended by the International Society for Cellular Therapy for identification of multipotent mesenchymal stromal cells (MMSCs), as well as the CD44 marker. Immunocytochemical staining of MSCs using specific monoclonal antibodies conjugated with fluorescein isothiocyanate revealed positive immunoreactivity of the isolated cells to CD 90, CD 105, CD 73, CD 44 and CD 13. The obtained data allowed us to conclude that the isolated cell preparation consists of mesenchymal stem cells and does not contain admixtures of hematopoietic stem cells.

(Minor Comments):

Add labels to the imaging figure panels so it’s easier to follow instead of reading the full caption (e.g. in Figure 2A:, label the top row Hoechst+, and the bottom row Propidium Iodide+ in the figure itself, so it’s quicker for the reader to follow, similar to how Figure 1f and g have labels beneath them).

In section 2.2, the authors states that there ‘was a dose-dependent’ response, but the 4 and 40x10^6 show very similar values. It looks almost like a stepped response, saturating at 80x10^6. Why do the authors think this is the case?

Indeed, the observed dose-dependent response in our study revealed a nonlinear (stepwise) pattern, with similar effects at 4×10⁶ and 40×10⁶ particles, followed by a more pronounced response at 80×10⁶, suggesting a possible saturation threshold. The low concentrations (4×10⁶ and 40×10⁶) may be below the critical threshold required to trigger a measurable functional response in hippocampal cells. The similarity in outcomes at these doses could indicate that only at 80×10⁶ particles does the sEV concentration reach a level sufficient to activate signaling pathways or mechanisms associated, directly or indirectly, with memory processes.

Moreover, the nearly identical responses at 4×10⁶ and 40×10⁶ may reflect the existence of two distinct mechanisms—one activated at lower doses and the other engaged at higher concentrations (80×10⁶). Although the response appears stepwise rather than linear, this does not contradict dose dependency.

For relevant plots, please include sample sizes (e.g. Fig 1C, Fig 2g etc.)

Thank you for the suggestion. Sample sizes were included.

Round 2

Reviewer 1 Report

Comments and Suggestions for Authors

The characterization experiment needed for this manuscript is not the traditional way for EV identity (NTA, WB for markers), but more comprehensive assays like miR-seq or proteomics. I understand that it's difficult to find the mechanism between specific molecules and function, at least providing some candidates will significantly improve the quality of this manuscript. Also, more discussion about possible mechanism should be added.

Author Response

Thank you for giving this comment and clarifying the previous question. We have characterized sEVs according to the position statement from the International Society for Extracellular Vesicles [23]. This is the basis for exosomes and other extracellular vesicle characterization. All other methods such as miR-seq or proteomics are currently under our study. Some data on miR-seq such as those shown by Luther et al. [doi: 10.1016/j.yjmcc.2018.04.012], Hayashi and Hoffman [doi: 10.1080/15476286.2017.1361098] and by other works [doi: 10.1016/j.drup.2025.101211; doi: 10.14218/JCTH.2022.00144], most are mostly related to cancer research. The most relevant to our research is the paper “Comparative Analyses of Human Exosome Proteomes” [doi: 10.1007/s10930-023-10100-0] where HEK293 cell line vesicle proteome was compared with literature data. We have included the summary of this interesting work in the discussion. Still, even this very extensive study has not presented any concrete mechanisms. We have included in the discussion their ideas and presented ours based only on our hypothesis. Please see all added paragraphs in the discussion section which is our attempt to answer your question.